# IGOR: Image-GOal Representations are the Atomic Control Units for Foundation Models in Embodied AI

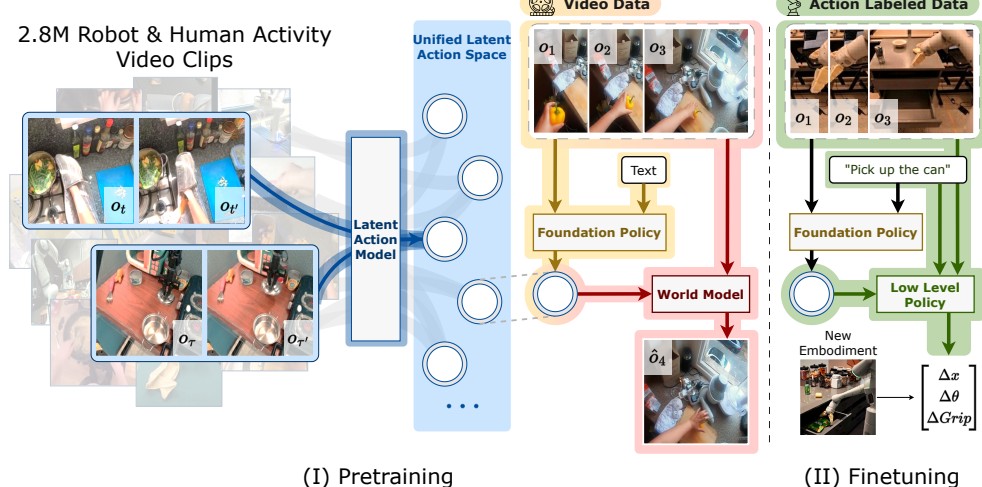

Figure 1: **Image-GOal Representations (IGOR) based training framework for embodied AI.** IGOR learns a unified latent action space for humans and robots by compressing visual changes between an image and its goal state on data from both robot and human activities. By labeling latent actions, IGOR facilitates the learning of foundation policy and world models from internet-scale human video data, covering a diverse range of embodied AI tasks. With a semantically consistent latent action space, IGOR enables human-to-robot generalization. The foundation policy model acts as a high-level controller at the latent action level, which is then integrated with a low-level policy to achieve effective robot control.

## Abstract

We introduce Image-GOal Representations (IGOR), aiming to learn a unified, semantically consistent action space across human and various robots. Through this unified latent action space, IGOR enables knowledge transfer among large-scale robot and human activity data. We achieve this by compressing visual changes between an initial image and its goal state into latent actions. IGOR allows us to generate latent action labels for internet-scale video data. This unified latent action space enables the training of foundation policy and world models across a wide variety of tasks performed by both robots and humans. We demonstrate that: (1) IGOR learns a semantically consistent action space for both human and robots, characterizing various possible motions of objects representing the physical interaction knowledge; (2) IGOR can "migrate" the movements of the object in the one video to other videos, even across human and robots, by jointly using the latent action model and world model; (3) IGOR can learn to align latent actions with natural language through the foundation policy model, and integrate latent actions with a low-level policy model to achieve effective robot control. We believe IGOR opens new possibilities for human-to-robot knowledge transfer and control. See video demonstrations on our anonymous webpage.

# 1 INTRODUCTION

Learning foundation models for embodied AI has been notably constrained by a lack of interaction data. Unlike text or video data, which are abundantly available, interaction data is much scarcer. Research efforts have been devoted to creating large-scale interaction datasets, such as Open-X-Embodiment (Collaboration et al., 2023) and DROID (Khazatsky et al., 2024). Based on multi-task interaction data, a series of generalist agents (or foundation policy models) have been proposed, such as RT-1 (Brohan et al., 2022), Robocat (Bousmalis et al., 2023), RT-2 (Brohan et al., 2023), Octo (Team et al., 2024), and OpenVLA (Kim et al., 2024). However, the volume of interaction data remains several orders of magnitude smaller than that of internet text or video data. Given that the success of foundation models relies on scaling up datasets and extracting knowledge from such large-scale datasets, it is essential to design methods for building embodied AI foundation models that can effectively utilize internet-scale video data.

Internet-scale video data contains abundant sequential records of human activities and perfect demonstrations of how human perform various tasks by interacting with the real world. When the human brain extracts information from videos, instead of doing it frame by frame, it modularizes the differences between frames into a single word such as "move", "open", "close". We refer to these highly compressed, modularized actions as latent actions that are shared across different tasks. The question to ask here is, **is it possible to recover latent actions from video datasets with humans and robots performing various real embodied AI tasks?** While recent works such as Genie (Bruce et al., 2024) and LAPO (Schmidt & Jiang, 2023) made attempts in recovering such latent actions from videos, they primarily focus on 2D platformer games where each latent action $a_t$ corresponds to a specific control button. The action space is highly designed to fit a specific scenario and incomparable to the complex human and robot action space in various embodied AI tasks. To take a step further, the question would be, **can we learn a unified, semantically consistent latent action space, allowing the transfer of knowledge across different tasks, and embodiments including human and various robots?**

In this paper, we propose Image-GOal Representations (IGOR), which learns a unified and semantically consistent latent action space shared across different tasks and embodiments, enabling the knowledge transfer among internet-scale video data. We propose a latent action model designed to capture robot and human actions across various embodied AI tasks. IGOR compresses the visual changes between an image and its goal state into latent actions, which are also embeddings of sub-tasks defined by reaching the goal from the initial image. IGOR is trained by minimizing the reconstruction loss of the goal state, which is predicted based on the image and the latent action. The core insight behind IGOR is that if compressed sufficiently, the image-goal pairs with similar visual changes will have similar embeddings.

> *We argue that, besides text embeddings for human instruction understanding and image/video embeddings for state understanding, image-goal representations for latent action learning and sub-task understanding are yet another crucial building blocks, which may hold great potential for becoming the atomic control unit in embodied AI.*

With the latent action model, we can transform internet-scale human video data into interaction data labeled with latent actions, which largely expands the data available to building embodied AI foundation models. This unified latent action space allows us to train foundation policy and world models on nearly arbitrary tasks performed by robots and humans. Specifically, we train a foundation policy model on large-scale video data with text labels. This model uses text to describe tasks and makes decisions, generating the next latent action to perform. Additionally, we train a foundation world model on the same dataset, learning to simulate the outcome of executing the foundation policy model. Image-Goal Representations can be viewed as atomic control units in visual space. They function both as latent actions for a foundational policy model to predict in visual trajectory planning and as sub-tasks for a robot-specific low-level policy to execute.

We train our models on human video data and robot data with actions removed, with RT-1 dataset held out for OOD evaluation. First, we evaluate the latent action model qualitatively, and find that image-goal pairs with similar latent actions have similar visual changes, corresponding to semantically consistent movements, even on OOD scenarios. Then we evaluate the world model by

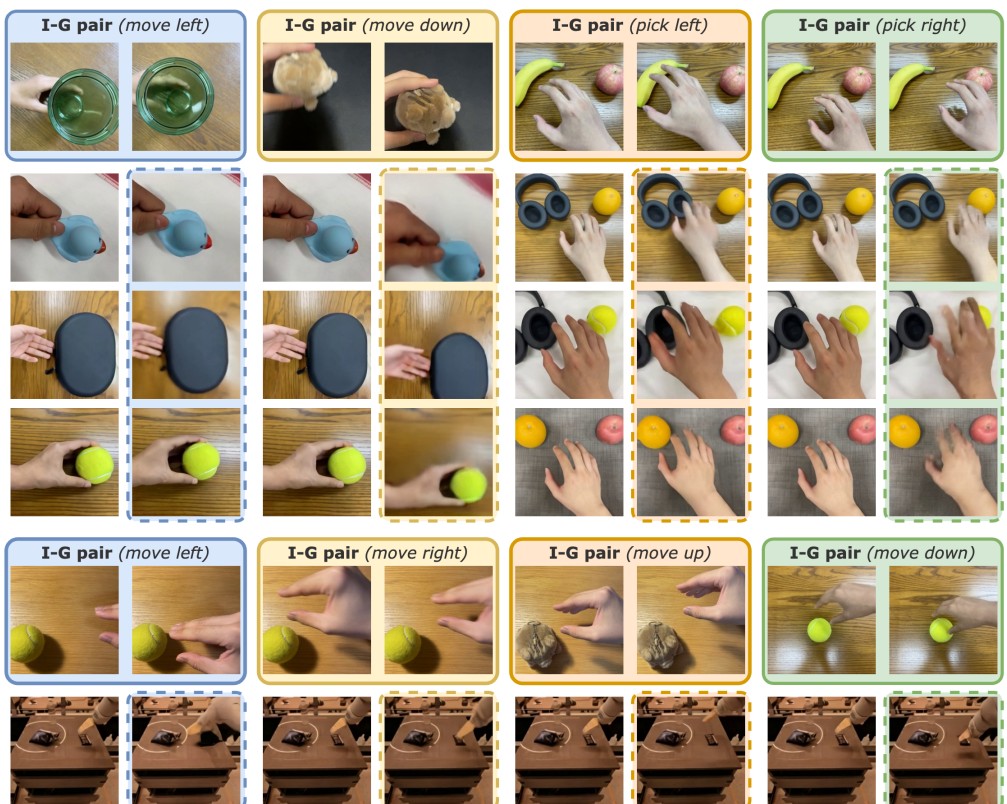

Figure 2: We extract latent actions from Image-Goal pairs in the solid line boxes, and apply the latent actions to different initial frames, generating subsequent videos via world model as shown in the corresponding dashed boxes. The first half illustrates examples from real-world videos with diverse object categories, while the second half demonstrates generalization from human to robot arms. Full videos are available on our website.

extracting latent actions from a video and applying such latent action (or action sequence) to the initial frames of other videos, generating the rest of frames. We find that, jointly with the latent action model and world model, IGOR successfully "migrates" the movements of the object in the one video to other videos, as shown in Figure 2. We also apply different latent actions to the same initial image, and find that the world model has learned various possible movements of the object in the image, suggesting that it has absorbed the physical interaction knowledge. For the foundation policy model, we show its ability in following diverse language instruction via iteratively rolling out the foundation policy and world model using latent actions. We further integrate it with a low-level policy, and show that IGOR-based policy training can improve performance on Google Robot tasks in low-data regime with the SIMPLER (Li et al., 2024) simulator.

## 2 METHODOLOGY

### 2.1 LATENT ACTION MODEL

The primary objective of the latent action model is to label latent actions from unlabeled open-domain videos in an unsupervised manner. Given a sequence of video frames $o_{1:t+1}$, the goal is to derive the latent action $a_t$, which captures the key information describing only the changes that occur at time step $t$, removing other redundant information. In contrast to prior works (Schmidt & Jiang, 2023; Bruce et al., 2024), which primarily focus on 2D platformer games where each latent action $a_t$ corresponds to a specific control button, we aim to develop a more generalizable model. Our model is designed to handle the significantly greater complexity of open-world scenarios, where

latent actions may not correspond to any specific underlying actions. This presents several additional challenges.

First, rather than focusing solely on absolute position of pixel changes, the latent action model must learn to capture semantic movements that remain consistent across varying scenarios. Moreover, due to the temporal redundancy, actions are often sparse given long contexts, which can lead the model to infer $o_{t+1}$ directly from the history, bypassing the need for a more informative latent action $a_t$.

To address these issues, we propose a novel model architecture. Our latent action model consists of a pair of Inverse Dynamics Model (IDM) and Forward Dynamics Model (FDM). IDM $I$ is trained to predict the latent action $a_t$ based on the full sequence of observations $o_{1:t+1}$. Instead of using the raw observations, we first apply random cropping $c_1$ to the inputs: $a_t = I(c_1[o_{1:t+1}])$. For the architecture of $I$, we first extract features for each frame through Vision Transformer (ViT) (Dosovitskiy et al., 2021) and then adopt a Spatio-Temporal transformer (ST-transformer) (Bruce et al., 2024; Xu et al., 2021) with a temporal causal mask as the backbone. Learnable readout tokens are then used to extract and compress the visual changes into $N$ tokens. To further compress the information stored in latent action, we apply vector quantization to each token, restricting them to a discrete codebook of size $|C|$. Finally, we derive the latent action $a_t \in \mathbf{R}^{N \times D}$ where $D$ is the dimension of each code. We refer to $a_t$ as the latent action embedding, or sub-task embedding, as they describe the information that takes the observation $o_t$ to the next observation $o_{t+1}$.

For the FDM $F$, we propose using a single-frame Vision Transformer to reconstruct $o_{t+1}$, in contrast to previous works (Schmidt & Jiang, 2023; Bruce et al., 2024), which reconstruct the next frame given the entire context $o_{1:t}$. This approach mitigates the case where the model might predict the next frame directly from the context, bypassing the latent action. By conditioning on a single frame, it encourages more information to flow into the latent action $a_t$. For reconstruction, we apply another random cropping $c_2$, and the next frame is predicted as $\hat{o}_{t+1} = F(c_2[o_t], a_t)$. By using different croppings $c_1$ and $c_2$, the model is encouraged to learn a more semantically invariant latent action across different trajectories. The models are trained jointly with the reconstruction loss $\|c_2[o_{t+1}] - \hat{o}_{t+1}\|^2$ and the commitment loss in vector quantization.

## 2.2 FOUNDATION WORLD MODEL

Our foundation world model is a continuous-time Rectified Flow (Liu et al., 2023b; Esser et al., 2024) that learns to predict the future frames $o_{t+1:T}$ conditioned on the history observation frame $o_{1:t}$, and future latent actions $a_{t:T-1}$. To achieve this goal, there are two key challenges: 1) Generating the photo-realistic frame that describes the states precisely; 2) Controlling the generated frames by the latent actions.

Accordingly, we start our foundation world model with the pre-trained Open-Sora (Zheng et al., 2024). It consists of two components: a 3D Variational AutoEncoder (VAE) that encodes the raw observation into latent space with $8 \times 8$ times downsampling in spatial dimension and $4\times$ times downsampling in temporal dimension; a Spatial-Temporal Rectified Flow Transformer (ST-RFT) that generates the latent from the text conditions. To enable the control from the observation and action, we make two modifications to the original Open-Sora: 1) We replace the original text input of the pre-trained model with our latent actions $a_{1:T}$ obtained from LAM. Zero-padding is applied for the last action. For each frame, we map the latent actions into a single token and feed it to the ST-RFT via the cross-attention mechanism; 2) We also make the generation conditioned on the output of FDM $\hat{o}_{t+1:T}$, which provides a coarse-grained prediction according to the input latent action. For the conditioning of $\hat{o}_{t+1:T}$, we encode it to the latent space with the same 3D VAE and directly add it to the noisy input element-wise.

Formally, Rectified Flow (Liu et al., 2023b; Albergo & Vanden-Eijnden, 2023; Esser et al., 2024) aims at directly regressing a vector field that generates a probability path between noise distribution and data distribution. For $n \in [0, 1]$, we define the interpolation between the two distributions as:

$$\boldsymbol{x}_n = (1-n)\boldsymbol{x}_0 + n\boldsymbol{x}_1, \tag{1}$$

where $x_0$ is the clean data, $x_1$ is the sampled noise, and $x_n$ is the noisy data. During training, we train a vector-valued neural network $\boldsymbol{x}_\theta$ with L2 loss:

$$\mathbb{E}_{n,\boldsymbol{x}_0,\boldsymbol{x}_1} \|\boldsymbol{x}_0 - \boldsymbol{x}_\theta(\boldsymbol{x}_n, n, a_{t:T-1}, \hat{o}_{t+1:T})\|^2. \tag{2}$$

Instead of predicting the conditional expectation directly, we follow Liu et al. (2023b) to parameterize the velocity with a neural network $\boldsymbol{v}_\theta$ and train it on:

$$L_{\text{world}}(\theta) = \mathbb{E}_{n,\boldsymbol{x}_0,\boldsymbol{x}_1} \| (\boldsymbol{x}_1 - \boldsymbol{x}_0) - \boldsymbol{v}_\theta(\boldsymbol{x}_n, n, a_{t:T-1}, \hat{o}_{t+1:T}) \|^2. \tag{3}$$

It should be noted that, our foundation world model can be fine-tuned to accommodate the different action spaces of robots with various embodiments. The fine-tuning of the foundational world model is left as future work.

## 2.3 Foundation Policy Model and Low-level Policy Model

The training of the policy model consists of two stages. In the first pretraining stage, taken as input the raw observation frames $o_{1:t}$ and a textual description $s$ for the task, the foundation policy model predicts latent actions $a_t = I([o_{1:t+1}])$ labeled by the IDM in the latent action model at each step. The training dataset of this stage is the same as that used for the latent action model, i.e., with large-scale and diverse sources of videos. In the second finetuning stage, we add an extra prediction component on the foundation policy model to predict real continuous robot actions, with taking the raw observations as well as the latent actions predicted by the first stage model as input. In this stage, only the prediction component (i.e., the low-level policy model) is optimized on small-scale and task-specific downstream datasets, while other components are frozen.

Specifically, similar to the latent action model, the backbone of foundation policy model is also a ST-transformer equipped with a ViT image encoder, with a feed-forward layer as the final prediction layer. The textual description $s$ is encoded to a latent representation by a pre-trained text encoder, which is concatenated with the observation representation encoded by the ViT encoder as the joint input to the model. We use the L2 distance between the predicted hidden output and the latent action as the loss function. Given a trajectory consists of $t$ observations $o_{1:t}$, the training objective can be written as:

$$L_{policy} = \| P([s; o_{1:t}]) - a_t \|^2, \tag{4}$$

where $P(\cdot)$ denotes the policy model.

In the second stage, we train the low-level policy model to predict the real continuous actions within each latent action, where the image-goal latent actions can be seen as representations for sub-tasks defined by reaching a goal from an initial image. The low-level policy model is also an ST-transformer with a prediction layer. The input consists of the textual representation $s$, the observation $o_{1:t}$ and latent actions predicted by the foundation policy model $P([s; o_{1:t}])$, which are concatenated together at the patch level as one part. The latent action $P([s; o_{1:t}])$ predicted by the foundation policy model also serves as sub-task embedding for the low-level policy model. We denote that each latent action corresponds to $\tau$ real robot actions, and the latent action $a_t$ corresponds to real robot action $u_t^{1:\tau}$. Denote the low-level policy model as $P_f(\cdot)$, we train the second stage model also by L2 distance:

$$L_{ft} = \| P_f([s; P([s; o_{1:t}]); o_{1:t}]) - u_t^{1:\tau} \|^2, \tag{5}$$

where only the parameters of the low-level policy are optimized.

# 3 Experiments

## 3.1 Dataset

In the pretraining stage, we construct a large-scale dataset comprising diverse domains, including robotic data from various embodiments and a substantial amount of human activity videos.

**Data Mixture.** For the robotic data, we select a subset of Open-X Embodiment dataset (Collaboration et al., 2023) with single arm end-effector control, excluding RT-1 dataset for out-of-distribution (OOD) evaluation. We follow the preprocessing and data mixture weights from Team et al. (2024); Kim et al. (2024). In total, we utilize approximately 0.8M robot trajectories. While our dataset includes data from real robots, we discard the associated actions and proprio-states, using only image frames and text instructions during pretraining. Additionally, we incorporate large-scale open-world videos with language instructions, including human daily activities from Something-Something v2

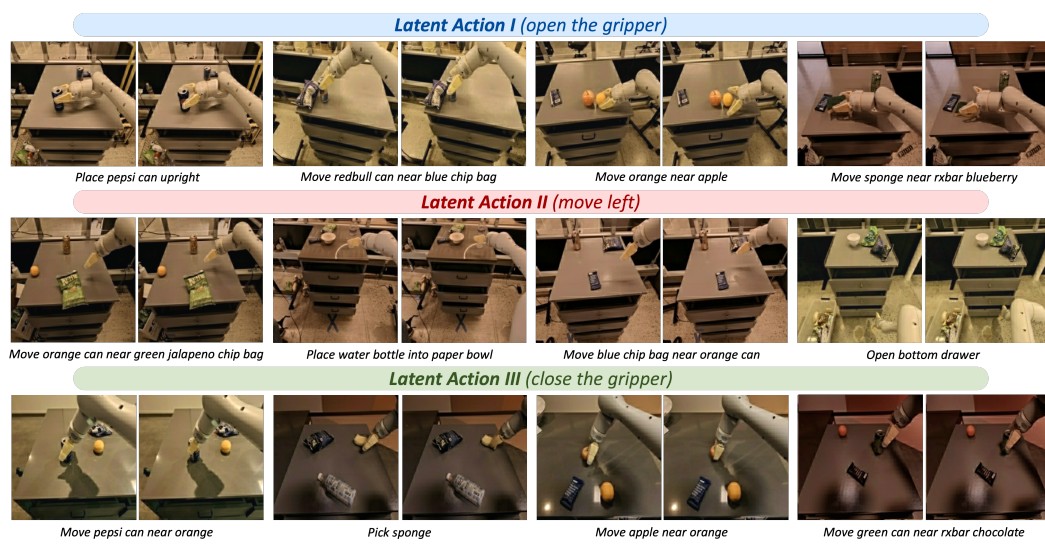

Figure 3: Image-goal pairs with similar latent actions in OOD RT-1 dataset. In each row, we choose the leftmost image-goal pair, and retrieve 3 nearest pairs on latent action embedding. The original task instructions of the pairs are shown under the images. We find that each row shares the similar visual changes semantically, and the latent actions generalize across different raw language tasks.

(Goyal et al., 2017), and egocentric videos such as EGTEA (Li et al., 2018), Epic Kitchen (Damen et al., 2020), and Ego4D (Grauman et al., 2022; Pramanick et al., 2023). In total, we derive approximately 2.0M human activity video clips with high quality. Overall, our dataset for pretraining comprises around 2.8M trajectories and video clips, where each trajectory contains a language instruction and a sequence of observations.

**Data Preprocessing.** In practice, we found that the video quality has a big impact on the model performance. We exclude low-quality videos characterized by excessive shakiness or rapid camera movement, and apply stabilization techniques to the remaining videos. To ensure proper amount of changes between frames in the latent action model, we choose the optimal frame rates for robotics dataset and human activity videos.

In the finetuning stage, we use RT-1 dataset, a large-scale dataset for real-world robotic experiences. We uniformly sample 1% number of episodes from RT-1 dataset for finetuning, where each episode comprises a language instruction, a sequence of image observations, and a sequence of low-level actions. The action space is 7-dimensional, including 3 dimensions of robot arm movement $\Delta Pos$, 3 dimensions of robot arm rotation $\Delta Rot$, and 1 dimension of robot gripper action $\Delta Grp$. We provide more details in Appendix A.

### 3.2 TRAINING DETAILS

We first pretrain our latent action model on our pretraining dataset. Then, we use the pretrained latent action model to label latent actions on our pretraining dataset, and pretrain foundation policy model and foundation world model on the labeled dataset. Finally, we finetune our low-level policy model on top of our pretrained models on RT-1.

For latent action model, we use a codebook with $N = 4$ tokens, and codebook size of $|C| = 32$, each with an embedding size of $D = 128$. We use a sub-task length of $\tau = 3$ for finetuning the low-level policy model on RT-1 dataset. Please refer to Appendix B for more training details.

### 3.3 QUALITATIVE RESULTS ON LATENT ACTIONS

We present qualitative results on latent actions learned from robotics and human activity dataset. Specifically, we answer the following questions on learned latent actions:

Figure 4: Controllability of latent action among multiple objects. The last two rows show the generated image by applying 6 different latent actions to the initial frame. Effects of applying different latent actions are highlighted in dashed squares: (a,b) move the apple, (c,d) move the tennis, (e,f) move the orange. Full generated videos from the world model are available on our webpage.

- Do similar latent actions reflect similar visual changes?
- Can latent actions encode semantically consistent changes across different tasks, and embodiments including human and robots? If so, are we able to migrate movements in videos across embodiments and tasks via latent action?
- Does the policy foundation model properly follow language instructions for task solving?

### 3.3.1 VISUALIZATION OF IMAGE-GOAL PAIRS WITH SIMILAR LATENT ACTIONS

We investigate whether similar learned latent actions reflect similar visual changes on robotics manipulation dataset. We use RT-1 dataset, which was excluded from the latent action model training and serves as out-of-domain samples for evaluation. We randomly select image-goal pairs from RT-1 dataset, and present the image-goal pairs with smallest euclidean distance in latent action embedding in RT-1 dataset in Figure 3. We observe that pairs with similar embeddings indeed have similar visual changes, and also similar sub-tasks in semantic, for example, "open the gripper", "move left", and "close the gripper". Furthermore, each sub-task appears in different raw language tasks, suggesting the latent actions are reused, thereby facilitating generalization in model learning.

### 3.3.2 CONTROLLABILITY OF LATENT ACTIONS

We demonstrate that latent actions are able to control the changes in objects on different real world scenes, and the effects of latent actions generalize across tasks and embodiments. Specially, the generalizability of latent actions enable IGOR to successfully migrate human movement videos into robot movements provided the initial image, despite they largely differ in embodiments.

**Object Controllability Among Multiple Objects.** We evaluate the controllability of the latent actions on object movements among multiple objects on the same image. In Figure 4, we generate subsequent images by applying 6 different actions to the same original image on the foundation world model. We observe that the latent action model and foundation world model learn to control specific object's movement among multiple objects.

**Object Controllability Across Embodiments and Tasks.** We evaluate the semantic consistency of the latent actions across different setups, including embodiments and tasks. We use pairs of image-goal in the real world manipulation videos to generate latent actions, and apply the same set of actions to other images in different scene setups with foundation world model to generate subsequent

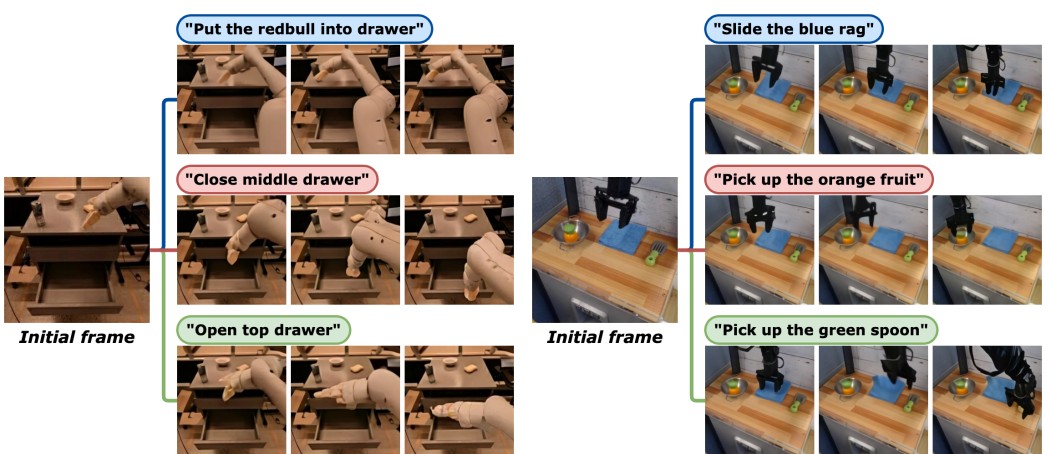

Figure 5: Generated image sequence jointly by the foundation policy and world model via only latent actions, following 3 different instructions from the same initial image. Full generated videos from the world model are available on our webpage.

videos. The results are shown in Figure 2. Impressively, we observe that (1) latent actions are semantically consistent across different tasks with different object categories; (2) latent actions are semantically consistent across human and robots. By applying latent actions extracted from human demonstrations, we generate videos of robot arm movements. With only one demonstration, the robot arm can successfully migrate human behaviors, which opens up new possibilities for few-shot human-to-robot transfer and control.

### 3.3.3 Counterfactual Video Generation with Diverse Instructions

We analyze whether the foundation policy model has the ability to follow human instructions. To this end, we interpret the effect of latent actions visually with the foundation world model. Starting from a single initial image, the foundation policy and world model can jointly generate diverse behaviors in videos that follow diverse instructions using only latent actions. We experiment with initial images from RT-1 and Bridge dataset and manually written instructions, and show the image clips of generated videos in Figure 5. The results show that the foundation policy model can properly follow different language instructions for task solving.

### 3.4 Quantitative Results

#### 3.4.1 Evaluation on the Google Robot Tasks in SIMPLER

We evaluate our IGOR-based training framework on the Google robot tasks in the SIMPLER simulator under a low-data regime, utilizing only 1% of the data from the large RT-1 dataset for the low-level policy learning stage.

**Evaluation Setups.** We test different model's ability to control the Google Robot following language tasks with RGB images as observations, where all robots are controlled with low-level end-effector control actions, after finetuning on the same amount of data from RT-1 dataset. We evaluate the success rate on three tasks: "Pick Coke Can", "Move Near", and "Open / Close Drawer".

**Baseline Method.** We compare our method with the same low-level policy model architecture with ST-Transformer, without latent action embedding concatenated on the observation feature embedding.

We present the success rate of different methods in Figure 6(a). From the figure, we observe that IGOR achieves higher or equal success rate than the model trained from scratch, showing the generalizability of the learned latent action to real robotics actions.

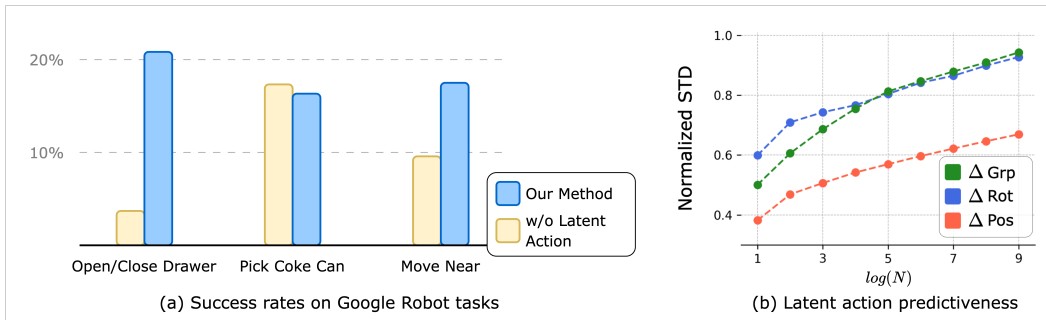

(a) Success rates on Google Robot tasks      (b) Latent action predictiveness

Figure 6: **(a).** Success rate of IGOR and the low-level policy trained from scratch methods on Google Robot tasks under SIMPLER simulator, finetuned on 1% data of RT-1. **(b).** Predictiveness of latent action on robot action. X-axis: $\log(N)$, where $N$ is the number of nearest neighbours in latent action embedding. Y-axis: normalized standard deviation in action embedding with respect to movement actions (orange), rotation actions (blue), and gripper actions (green).

### 3.4.2 PREDICTIVENESS OF LATENT ACTIONS ON ROBOT ACTIONS

We analyze whether our learned latent actions are predictive of real robot actions. On RT-1 dataset, we randomly sample a number of $M = 15,000$ pairs of images, and compute their latent action embeddings. For each pair of image, we find $N$ nearest neighbours of image pairs in RT-1 dataset with the closest latent action embedding, and compute the standard deviation of real robot actions among $N$ neighbours on each action dimension, normalized by the standard deviation of robot actions over each dimension over the whole RT-1 dataset. By varying $N$, we assess whether closer latent actions correspond to more similar downstream actions.

The results are shown in Figure 6(b). The fact that smaller $N$ leading to lower normalized standard deviation, and all normalized standard deviation being below $1.0$, show that the latent actions are predictive of real robot actions including robot movements, rotations and gripper actions. It is also shown that the latent actions are more predictive of the robot movement than rotations and gripper actions, suggesting that the IGOR learned action space reflects more information in robot movements than robot arm rotations and gripping.

### 3.5 ABLATION STUDIES

We provide additional ablation studies on the pretraining dataset of latent action model, showing that using a mixture of robotics and human activity dataset benefits the generalization of latent action model. Detailed ablation studies results are provided in Appendix C.

## 4 RELATED WORK

**Foundation Agents for Robots** Open-ended task-agnostic training and high-capacity neural network architectures have been recognized as key to the success of foundation models. In this context, a series of generalist agents have been proposed as the foundation policy models for robots (Brohan et al., 2022; Bousmalis et al., 2023; Brohan et al., 2023; Team et al., 2024; Kim et al., 2024). RT-1 (Brohan et al., 2022) contributes a large-scale multi-task dataset and a robotic transformer architecture, facilitating and assessing generalization across multiple tasks. RoboCat builds on Gato (Reed et al., 2022), further enabling multi-embodiment generalization. RT-2 highlights the importance of leveraging vision-language models trained on internet-scale data (Brohan et al., 2023). Octo (Team et al., 2024) and OpenVLA (Kim et al., 2024) can be seen as open versions of RoboCat and RT-2 respectively, with some additional technical contributions. IGOR is similar to RT-2 and OpenVLA in the sense that we both leverage Internet-scale data. The difference lies in that we use video data of human/robot performing embodied AI tasks, while they use text data and visual question answering data for the training of vision language models. To the best of our knowledge, we present the first foundation policy model that performs decision making at the sub-task (i.e. latent action) level.

**Image-Goal Visual Changes** Tracking visual changes and establishing correspondence between an image and its goal state is crucial for dynamic visual understanding in embodied AI. SiamMAE (Gupta et al., 2023) proposes to use a siamese encoder on the image and goal to learn visual correspondence. Voltron (Karamcheti et al., 2023) introduces language-guided visual represen-

tation learning on image-goal pairs. Lin et al. (2024) and Ko et al. (2023) leverage optical flow between image and goal to capture visual changes and correspondence, while Video-LaVIT (Jin et al., 2024) utilizes motion vectors. iVideoGPT (Wu et al., 2024) proposes using image-conditioned goal representations as state representations to predict within a world model. VPT (Baker et al., 2022) proposes to recover latent actions in videos using an inverse dynamics model trained on interaction data to predict real actions. Perhaps the most similar approaches to our methods are LAPO (Schmidt & Jiang, 2023) and Genie (Bruce et al., 2024). Both works primarily focus on 2D platformer games where each latent action corresponds to a specific control button. By contrast, we aim to develop a more generalizable model to handle the significantly greater complexity of open-world scenarios, where latent actions may not correspond to any specific underlying actions.

**Video Generation for Embodied AI**   Video generation is another research topic closely related to embodied AI. It has been proposed that video can be seen as the new language for real-world decision making (Yang et al., 2024b). Many works on world models build on video generation techniques (Bruce et al., 2024; Wu et al., 2024; Hu et al., 2023; Yang et al., 2024a; Xiang et al., 2024). Some text-to-video works claim to be real-world simulators, such as Sora (Brooks et al., 2024) and WorldDreamer (Wang et al., 2024). Unipi (Du et al., 2023) proposes to first predict the next goal state, then infer real robot actions with an inverse dynamics model. By contrast, our foundation policy model first predicts the latent action, which can specify the goal state, and then uses the latent action to enable sub-task level generalization. We argue that forward prediction in latent action space, rather than the original image space, offers several advantages. For example, we can perform sub-task understanding for image-goal representations, and the compressed latent action could be easier to predict than the entire image.

**Pre-trained Visual Representations**   Pre-trained Visual Representations target on training representations for images/videos in self-supervised learning manner (He et al., 2021; Xiao et al., 2022; Radosavovic et al., 2022; Majumdar et al., 2023; Radford et al., 2021; Nair et al., 2022; Ma et al., 2023; Oquab et al., 2023; Darcet et al., 2023; Kirillov et al., 2023; Assran et al., 2023; Bardes et al., 2024), and has been demonstrated to be very effective for state understanding in embodied AI. By contrast, IGOR learns image-goal representations for sub-task understanding, which we believe are another crucial building blocks, that may significantly enhance generalization in embodied AI.

## 5   CONCLUSIONS, LIMITATIONS, AND FUTURE WORK

In this paper, we propose IGOR, a novel training framework, taking the first step towards learning a unified action space for humans and robots in various embodied AI tasks.

Qualitatively, we demonstrate that:

- IGOR learns similar representations for image pairs with similar visual changes.
- The learned latent action has control over the next state given the current image.
- The foundation world model acquires knowledge about objects and their potential movements.
- The foundation policy model learns to follow instructions across different states.

Quantitatively, we show that:

- On the RT-1 dataset, image-goal pairs with similar latent actions have similar low-level robot actions.
- The IGOR framework improves policy learning, potentially due to its capability to predict the next sub-task by leveraging internet-scale data, thereby enabling sub-task level generalization.

The IGOR framework is limited in the following perspective: we cannot separate visual changes caused by the agents, other agents (such as dogs), or the shakiness of the camera. To address this, we mitigated camera shakiness and used only ego-centric videos without other agents in view. Just like any other representation learning methods, scaling up the dataset and model size is always most straightforward and effective. To facilitate the usage of more data, incorporating image processing methods such as object segmentation with IGOR will be part of future works. For better applications in embodied AI, the foundation world model can also be tuned to match real world scenarios, along with other improvements such as adapting the latent action model for multi-agent scenarios.

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

# A  DATASET

We present the datasets used for pre-training in Table 1. In total, these datasets comprise approximately 0.8 million robot trajectories and 2.0 million filtered human activity video clips. The robot data ratios are from (Team et al., 2024).

| Robot Dataset | Mix Ratio (%) |
|---|---|
| Kuka (Kalashnikov et al., 2018) | 7.72 |
| Bridge (Walke et al., 2023; Ebert et al., 2021) | 8.08 |
| Taco Play (Rosete-Beas et al., 2022; Mees et al., 2023) | 1.82 |
| Jaco Play (Dass et al., 2023) | 0.24 |
| Berkeley Cable Routing (Luo et al., 2023) | 0.12 |
| Roboturk (Mandlekar et al., 2018) | 1.40 |
| Viola (Zhu et al., 2023b) | 0.55 |
| Berkely Autolab UR5 (Chen et al.) | 0.73 |
| Toto (Zhou et al., 2023) | 1.21 |
| Language Table (Lynch et al., 2023) | 2.67 |
| Stanford Hydra Dataset (Belkhale et al., 2023) | 2.67 |
| Austin Buds Dataset (Zhu et al., 2022) | 0.12 |
| NYU Franka Play Dataset (Cui et al., 2022) | 0.49 |
| Furniture Bench Dataset (Heo et al., 2023) | 1.46 |
| UCSD Kitchen Dataset (Yan et al., 2023) | 0.06 |
| Austin Sailor Dataset (Nasiriany et al., 2022) | 1.34 |
| Austin Sirius Dataset (Liu et al., 2023a) | 1.03 |
| DLR EDAN Shared Control (Quere et al., 2020) | 0.06 |
| IAMLab CMU Pickup Insert (Saxena et al., 2023) | 0.55 |
| UTAustin Mutex (Shah et al., 2023) | 1.34 |
| Berkeley Fanuc Manipulation (Zhu et al., 2023a) | 0.43 |
| CMU Stretch (Mendonca et al., 2023) | 0.12 |
| BC-Z (Jang et al., 2022) | 4.56 |
| FMB Dataset (Luo et al., 2024) | 4.31 |
| DobbE (Shafiullah et al., 2023) | 0.85 |
| DROID (Khazatsky et al., 2024) | 6.07 |
| Ego4D (Grauman et al., 2022) | 32.1 |
| Something-Something V2 (Goyal et al., 2017) | 9.5 |
| EPIC-KITCHENS (Damen et al., 2020) | 8.0 |
| EGTEA Gaze+ (Li et al., 2018) | 0.4 |

Table 1: Dataset, mixture weights, and number of training examples after filtering in the pre-training stage in IGOR.

**Data Filtering**  We observed that video quality significantly affects the action model, particularly for human activities video. Excessive shakiness in videos can introduce visual changes between consecutive frames that are unrelated to the agent's actions.

We calculate the camera motion over the videos, and filter approximately 40% percent of open-world video data. For the remaining data, we further stabilize the videos. Although we retain only 60% percent of open-world video data, we find that the action model improves dramatically.

**Frame Interval**  A noticeable amount of visual changes is crucial for our latent action model. If we select two frames that are too close in time, the agent may barely move, resulting in visual changes that are not significant enough for inferring meaningful actions. Conversely, if the frames are too far apart, the changes might be too large to model accurately. To address this issue, we tune the sampling interval. For robot data, we choose frames that are three intervals apart, using $s_t$ and $s_{t+3}$ as the image-goal pair. For real world videos, we control the sampling. For real world data, we control the sample interval to be within $[0.1s, 0.5s]$. For the action and policy model, the context frames follow the same interval, ensuring that each pair of consecutive frames maintains this consistent spacing.

## B  TRAINING DETAILS

### B.1  LATENT ACTION MODEL TRAINING

The latent action model uses an ST-transformer equipped with a frozen DINO-v2 pretrained ViT image encoder. The latent action model uses 258 M parameters, a patch size of 14, and a codebook with $N = 4$ tokens and size $|C| = 32$, each with an embedding size of $D = 128$. We train the latent action model with batch size $B = 512$, training iterations of 140K steps, and learning rate of $1.5e - 4$ with Adam optimizer.

### B.2  FOUNDATION WORLD MODEL TRAINING

We start on the top of the OpenSora (Zheng et al., 2024) model with newly initialized projection layers. The foundation world model with batch size $B = 12$, training iterations of 48K, and learning rate of $1e - 4$ with Adam optimizer.

### B.3  FOUNDATION POLICY MODEL AND LOW-LEVEL POLICY MODEL

The latent action model uses an ST-transformer equipped with a frozen DINO-v2 pretrained ViT image encoder, following the latent action model's image encoder. The foundation policy model consists of 12 layers of spatial and temporal attentions, each with 12 attention heads and hidden dimension as 768 and a patch size of 14. In total the policy model has 138M parameters. We use frozen CLIP features for text instructions. We pretrain the foundation policy model with batch size $B = 128$, training iterations of 124K, and learning rate of $1e - 4$ with Adam optimizer.

The low-level policy model adds extra 118M parameters on top of the foundation policy model. We use a sub-task length of $\tau = 3$ for finetuning the low-level policy model on RT-1 dataset. We finetune the low-level policy model with batch size $B = 128$, training iterations of 32K, and learning rate of $1e - 4$ with Adam optimizer.

## C  ADDITIONAL ABLATION RESULTS

### C.1  DATASET ABLATION FOR LATENT ACTION MODEL

We compare two different settings for the pre-training dataset: only use the robotic dataset (robot data), and use both robotic and human activity dataset (mixed data). We evaluate the validation loss on the latent action model on RT-1 dataset, which is held out from the pretraining dataset and serves for OOD evaluation. Validation loss of the latent action model assesses the extent to which the IDM and FDM can jointly generate latent actions and recover goal states from these latent actions conditioned on states on the unseen dataset. The results are shown in Table 2. We find that the OOD validation loss is greatly reduced by adding human activity dataset. This may be due to the diversity of human videos, which comprise real daily life environments with lots of diverse backgrounds and objects. These results demonstrate that it is promising to leverage human data for improving robot tasks under the IGOR framework.

|  | Validation Loss |
|---|---|
| Robot Data | 0.145 |
| Mixed Data | 0.112 |

Table 2: Validation loss on held-out dataset (RT-1) with different training data.

## D  MODEL STRUCTURE OF IGOR

We illustrate the model architecture of latent action model, policy model and world model in IGOR in Figure 7.

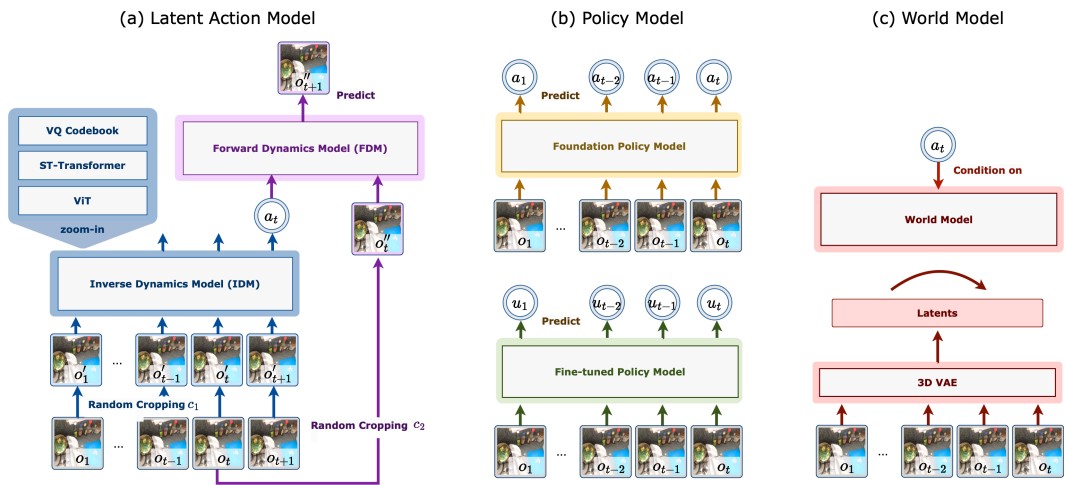

Figure 7: **The Model Structure of IGOR.**

# E ADDITIONAL DISCUSSION ON RELATED WORK

Any-Point Trajectory Modeling (ATM) (Wen et al., 2023) and Tract2Act (Bharadhwaj et al., 2024c) generates point tracking from the action-free videos for pretraining the policy to predict future trajectories of the tracked point, and learns a robot policy conditioned on generated trajectories. HOPMan (Bharadhwaj et al., 2024b) generates future human plans as conditions for facilitating robot policy learning. Im2Flow2Act (Xu et al., 2024) conditions the robot policy on complete generated object flows, which captures movement information only for the object, improving its cross-embodiment transfer capability.

Same as ATM, HOPMan, Im2Flow2Act, and Tract2Act, IGOR also pretrain a foundation policy on action-free videos that generalize across embodiments. There are two major differences: (1) IGOR uses an unsupervised way to learn and compress the visual changes, while existing work needs an additional pretrained video model for point tracking; (2) IGOR uses a compact latent representation for visual changes, while existing works uses an explicit representation for visual changes. The compact action representation enables IGOR to transfer human actions on robots directly, as shown in Figure 2 in our paper.

In the world model aspect, Gen2Act (Bharadhwaj et al., 2024a) uses the VideoPoet model (Kondratyuk et al., 2023) for text-to-video generation, and the generated human videos are used for downstream policy learning. Compared to existing world model works, IGOR can learn generalizable latent actions for fine-grained control on manipulation scenarios for both human and robots, while existing works use coarse grained text conditions for generating video predictions.

