# OpenReview forum: "IGOR: Image-GOal Representations are the Atomic Building Blocks for Next-Level Generalization in Embodied AI"
_ICLR.cc/2025/Conference — Submitted to ICLR 2025_

### Official Review · Reviewer_agiQ · 2024-10-26

**Soundness:** 3
**Presentation:** 3
**Contribution:** 3
**Rating:** 6
**Confidence:** 4

**Summary:**

The paper proposes Image-GOal Representations (IGOR), which learns a unified and semantically consistent latent action space shared across different tasks and embodiments, enabling the knowledge transfer among internet-scale video data. The authors show the efficacy of the proposed approach by providing both qualitative and quantitative results showcasing the generalizability of IGOR for different tasks and across different modalities. The paper also includes robot policy learning results to highlight the effectiveness of pretraining with latent actions for robot learning.

**Strengths:**

- The paper proposes IGOR, a method that learns a shared semantically consistent latent action space across different tasks and embodiments, enabling the knowledge transfer among internet-scale video data.
- The paper proposes a novel architecture for learning a latent action model and combines it with a foundational policy and world model learned on this latent action space. Further, a low level policy can be learned to map this latent action to a robot action making the learned model useful for robot learning.
- The paper includes qualitative results showing that the same latent action applied to different scenes and embodiments leads to similar future frames, highlighting the generalizability of the learned latent action space.
- Quantitative results showing improvements in policy performance (Fig 6a) and predictiveness of the latent action for robot actions (Fig 6b)  further emphasize the usefulness of such a system for robot control.

**Weaknesses:**

- The paper proposes a latent action model architecture combining inverse and forward dynamics modeling. Based on the description in the paper, the inverse model architecture seems very specific using ViT for patch processing followed by a spatio-temporal transformer. Then vector quantization is added to each readout token followed by the latent action being derived from these vector quantized tokens. Ablation studies justifying these design choices would be great and would help the community get insights about architectural choices that matter when designing such models.
- What are the resources and time required to train this model? And what is the inference frequency when running the policy?
- From Fig 6b, it seems like the latent actions are more predictive of robot positions than rotations or gripper movements. The tasks shown in the paper include opening/closing a drawer, picking up a coke can, and moving near which I do not think require much rotations. Since the goal of this work is to show that such a unified latent action helps embodied AI, it would be great if the authors could include more diverse tasks with varying levels of difficulty.
- Further, I would be curious to see if the proposed latent action struggles when a task involves larger rotations (since this would be a potential limitation of the method).

**Questions:**

It would be great if the authors could address the questions/concerns mentioned in Weaknesses. I am willing to increase my score once these concerns have been addressed.

---

> ### Author Response · Authors · 2024-11-22
> **Response**
>
> We would like to thank the reviewer for valuable feedback and for the comments and questions for additional clarity. Below, we address the key comments and questions.
>
> ---
>
> **W1. Rationale and ablations on the architecture design**
>
> Thank you for your suggestion! To control training costs, we conducted ablation studies on a small subset of the training data and provided visualization results using FDM in the action tokenizer.
>
> **a) Latent Action Extraction.**
>
> We compared two approaches: (1) applying Conv2D on re-organized image patches, and (2) using readout tokens. With the same latent dimension, readout tokens ( shown in https://raw.githubusercontent.com/iclr-2025-4517/index/refs/heads/main/images/rebuttal_imgs/rebuttal_fig2.png ) achieved better reconstruction results than Conv2D ( https://raw.githubusercontent.com/iclr-2025-4517/index/refs/heads/main/images/rebuttal_imgs/rebuttal_fig1.png ), suggesting it help extract information more efficiently.
>
> **b) Leveraging Temporal Information.**
>
> We compared two methods: (1) using a vanilla ViT on each frame, and extracting information between the adjacent patches, and (2) adding an ST transformer after the ViT to fuse more history context. As shown in figure [ https://raw.githubusercontent.com/iclr-2025-4517/index/refs/heads/main/images/rebuttal_imgs/rebuttal_fig3.png ] the pure ViT version tended to directly copy the robot arm's position instead of learning relative changes. In contrast, adding historical information mitigated this issue. We hypothesize that incorporating historical context helps the action tokenizer to identify key objects changes. Additionally, applying random cropping, as discussed in Section 2.1, further improved the results.
>
> We hope our findings and trials may bring some insights to the community!
>
> ---
>
> **W2. The training resources and inference frequency**
>
> For action tokenizer training, we used about 200 H100 GPU days. For policy and world model, we each used about 100 H100 GPU days to train.
>
> Regarding inference frequency, we tested the policy's inference speed by performing 1,000 forward passes on a local A100 GPU. The process took approximately 40.2 seconds to complete, suggesting the policy itself can run at roughly 25Hz.
>
> ---
>
> **W3. Regarding the rotation information in latent actions**
>
> Thank you for your suggestion! While Fig. 6b shows that latent actions are more predictive of positions, we argue that they still retain sufficient information about rotations.
>
> We acknowledge that the tasks in Simpler environments are limited. To address this, we propose an experiment where a video of a rotating and moving robot gripper is encoded into a latent action sequence. This sequence is then applied to the first frame and decoded back into a video through the world model.
>
> If the latent actions contain rotation information, the reconstruction will accurately reflect both rotation and movement. Otherwise, the reconstruction will only show movement.
>
> The results are available on “Latent Action for Rotation” part of the website [ https://iclr-2025-4517.github.io/index/rebuttal.html ], showing the original video on the left and the reconstructed video on the right. From the video, it is evident that the robot arm both rotates and moves. Though the gripper is slightly distorted when rotating, it completes the full trajectories with both rotation and movement, suggesting the latent action captures enough rotation information.

---

> ### Author Response · Authors · 2024-11-26
> **We hope that our response addresses your concern**
>
> We sincerely appreciate the time you have taken to review our response. With our rebuttal now submitted, we are eager to know if it has adequately addressed your concerns. As the rebuttal phase draws to a close, we would be grateful for any additional feedback or clarification you might require.

---

### Official Review · Reviewer_dXPY · 2024-11-04

**Soundness:** 3
**Presentation:** 3
**Contribution:** 3
**Rating:** 6
**Confidence:** 4

**Summary:**

Igor defines a unified latent action space, with the goal of enabling transfer among large-scale robot and human activity data.    Visual changes between an initial image and its goal state are compressed into latent actions from internet-scale video data.  IGOR learns a semantically consistent action space for humans and robots, can ‘migrate’ the movements of the object in one video to other videos and can learn to align latent actions with natural language through the foundation policy model.  IGOR is trained by minimizing the reconstruction loss of the goal state.  If compressed sufficiently the image-goal pairs with similar visual changes will have similar embeddings.  It is trained on human video data and robot data with the actions removed.  It has been integrated with a low-level policy and show improved perfomance in low data regime and on SIMPLER.

The latent action model consists of a pair of IDM and FDM.  IDM is trained to predict a given the full sequence of observations, vision features are extracted from a ViTa nd then an ST-Transformer with a temporal causal mask.  Learnable readout tokens are used to extract and compress the visual changes.  VQ is applied to each token.  a_t is the latent action embedding.  FDM uses a single frame ViT to reconstruct o_t+1 with reconstruction loss.

The world model is continuous time rectified flow, and learns to predict future frames conditioned on the history and the future latent actions.  Open-SORA is the starting point: a 3D VAE that encodes to a latent space and an ST-RFT that generates the latent from text conditioning.  Two modificaitons were made: the text input is replaced with latent actions and the generation is conditioned on the output of the FDM (2:T).

The foundation policy model and low-level policy.    For pretraining, raw observations input and latent actions are output.  Training is the same as the LAM.  The second fine-tuning stage a low-level policy is added to generate continuous outputs.  It’s an ST-transformer with ViT image encoder.  Low-level is an ST-transformer.

Section 3 describes the experiments.  Data comes from open-x, ssv2, EGTEA, EGO4D.  Filtering is done on video quality.  Finetuning is on RT1.  LAM is pre-trained first.  The pretrained LAM labels is used to label actions on the pretraining dataset and to retrain the foundation policy and world model.  The low level p[olicy is fine-tuned on top of the RT-1 policy.  Image-goal pairs are visualized in Fig. 3, latent actions are able to control the changes in objects in different real-world scenes.  LA are semantically consistent across different tasks and human / robot.  Foundation policy can properly follow different language instructions.  Quantitative results in SIMPLER and on the predictiveness of latent actions on robot actions.

**Strengths:**

Overall a great paper that I’ve been wanting to see for a while (e.g., latent actions for pretraining); it’s well-written, detailed, should be reproducible and innovative.

**Weaknesses:**

The only downsides to the paper are 1) that the evaluation is only on a few tasks and in SIMPLER (the promise of these approaches is more generalization, so it would be great if there were some real robot experiments) 2) that the pretraining dataset is on the smallish side for foundational capabilities.  It is great that the authors can show such capabilities at a small scale as a proof of concept even still.  3) demonstration of broader foundational capabilities (related to point 1).

**Questions:**

* It would be great if you could add / share the intuition for adding why having an IDM and FDM?

p.8 image -> pair of images

---

> ### Author Response · Authors · 2024-11-22
> **Response**
>
> We really appreciate reviewer dXPY for the positive and encouraging feedback!
>
> **The intuition on adding IDM and FDM.**
>
> We would love to share more intuition on the IDM and FDM for the latent action model. The key idea is based on the observation that we can easily identify the object correspondence between the image and goal (when they are not too far apart in time) and infer the motion of objects/arms solely from visual changes. Based on this key idea, the IDM takes a pair of image and goal (as well as historical frames) as input, allowing it to infer the visual changes by comparing the current image with the goal. The FDM then takes the current image and the latent action as input to reconstruct the goal as the training loss, which ensures that the latent action contains all the information about the visual changes. By compressing the latent action effectively, image-goal pairs with similar visual changes will have similar latent action embeddings, as demonstrated in the experiments and video demos presented in our paper.

---

> > ### Comment · Reviewer_dXPY · 2024-11-29
> > **Response to Authors**
> >
> > After reading through the other reviewer's comments:
> > * Experiments: I would like to acknowledge that additional results beyond the SIMPLER benchmark and specifically on real robots would make the paper substantially stronger.
> > * Latent Action Representation: A comparison of latent action representations (as the first reviewer mentioned) would also make the paper substantially stronger, though perhaps isn't required for publication.  At a minimum a robust discussion in related work would be required.
> >
> > Overall, while the paper could be stronger, at the same time I do think in it's current form it would be an interesting paper to have at the conference.

---

> ### Author Response · Authors · 2024-11-26
> **We hope that our response addresses your concern**
>
> We sincerely appreciate the time you have taken to review our response. With our rebuttal now submitted, we are eager to know if it has adequately addressed your concerns. As the rebuttal phase draws to a close, we would be grateful for any additional feedback or clarification you might require.

---

### Official Review · Reviewer_68vo · 2024-11-05

**Soundness:** 2
**Presentation:** 2
**Contribution:** 2
**Rating:** 3
**Confidence:** 4

**Summary:**

This paper proposes a method to leverage human videos for training a policy and world model in a latent-action space. They are then combined with a low-level policy trained on robotic data to perform downstream manipulation tasks. Qualitative results demonstrate that the model effectively learns latent actions, and the use of latent actions is shown to improve downstream performance on SIMPLER tasks.

**Strengths:**

The motivation behind using human videos to learn a latent-action model is both interesting and important. The qualitative results and videos demonstrate that the proposed method can effectively learn a reasonable latent action space.

**Weaknesses:**

1. [Major] The major concern of this paper is the lack of quantitative results and comparisons. The latent action model is primarily presented through qualitative visualizations, and the downstream performance is only compared to a variant without latent action conditioning. It should also be compared to other approaches. For instance, it’s worth comparing to methods like UniPi [1] and SuSIE [2] which utilize video/subgoal generation combined with low-level robot action inference. These methods have a similar assumption to this paper, as they also use internet-scale video data pre-training and a small amount of robot data.

2. [Major] It is also worth designing and evaluating the downstream robotic tasks from a more comprehensive perspective regarding generalization. The three tasks in SIMPLER used in the paper are in-distribution tasks included in the RT-1 dataset, and there is no evaluation of unseen tasks or more out-of-distribution tasks.

3. [Major] The paper makes several claims that are not convincingly supported by the experiments (see below).

> “l.93: which may hold great potential for next-level generalization in embodied AI.”

There is no experimental evidence demonstrating that the proposed method achieves 'next-level' generalization. It is not compared to any other zero-shot manipulation methods, as mentioned earlier. Additionally, it is unclear what is meant by 'next-level’. The same question applies to the paper’s title.

> “l.502: We argue that forward prediction in latent action space, rather than the original image space, offers several advantages. For example, we can perform sub-task understanding for image-goal representations, and the compressed latent action could be easier to predict than the entire image.”

There is no evidence to support this argument. There are no comparisons to UniPi kind of approach as mentioned above.

4. [Minor] The use of the term 'foundation' in 'foundation policy/world model' seems a bit arbitrary. What does it specifically mean?

---
Reference

[1] Du et al., Learning Universal Policies via Text-Guided Video Generation, 2023.

[2] Black et al., Zero-Shot Robotic Manipulation with Pretrained Image-Editing Diffusion Models, 2024.

**Questions:**

1. Could the authors provide more details on the data preprocessing? Specifically, how do you exclude low-quality videos, what are the 'stabilization techniques' mentioned in the paper, and what is the reasoning behind the chosen optimal frame rate?


2. It is unclear what is the purpose of the world model? It is entirely not used for low-level control.

---

> ### Author Response · Authors · 2024-11-22
> **Response**
>
> We would like to thank reviewer 68vo for recognizing the importance of our latent action approach. Below, we address the key comments and questions.
>
> ---
>
> **W1. Missing baselines: UniPi**
>
> Thank you for pointing out potential baselines. We implemented UniPi as our baseline. For the video generation model, we fine-tuned the pretrained OpenSora which is the same backbone as our world model. Additionally, we trained an inverse dynamics model to predict real robot action of google robot in Simpler environment.
>
> The success rates are as follows, demonstrating that our method achieves higher success than UniPi rates across all tasks.
>
>
> |            | Open/Close Drawer | Pick Coke Can | Move Near |
> |------------|-------------------|---------------|-----------|
> | Scratch    |        3.7        |      17.3     |    9.6    |
> | UniPi      |        4.6        |      1.6      |     5     |
> | Our Method |        20.8       |      16.3     |    17.5   |
>
> ---
>
> **W2. Expect experiments on more tasks**
>
> We acknowledge that the tasks in Simpler environments are limited. Currently, we do not have real world robots for real world tasks. At this stage, our method focuses on verifying the feasibility of the latent action approach. In future work, we will explore its generalization across a broader range of tasks and real world robots.
>
>
> ---
>
> **W3. several claims that are not convincingly supported by the experiments**
>
> Thank you for your valuable suggestions regarding our wording. We greatly appreciate your feedback and have revised our claims to be more precise and rigorous. Based on your input, we have updated the title to: “IGOR: Image-GOal Representations are the Atomic Control Units for Foundation Models in Embodied AI”. Here, "control units" emphasize the role of latent actions in enabling the learning of both foundation policy models and foundation world models from video data. The main paper will also be updated accordingly.
>
> Regarding the claim about the latent action space versus the original image space, we have implemented UniPi as our baseline, as discussed in W1, and we hope this addition will support our argument.
>
> By designing a unified latent action space, the IGOR framework facilitates the learning of foundation policy and world models using both robotic data and internet-scale videos. While the current dataset may not yet be large enough to serve as a foundation model, the method is fully scalable. We plan to expand the dataset to further improve the learning of foundation models in future work.
>
> ---
>
> **Q1. How is the data preprocessed? Why choose the optimal frame rate?**
>
> For data preprocessing, we employed a combination of automated and manual filtering and stabilization. Automated methods included a shaking-detection model and video stabilization software to enhance data quality, while manual inspection addressed edge cases.
>
> The optimal frame rate was selected to ensure latent actions capture meaningful visual changes. We choose interval equals to 3 for OpenX dataset, and the time interval to be in [0.1, 0.5] for egocentric videos. A frame interval that is too short results in nearly identical frames, where latent actions represent minimal changes. Conversely, a frame interval that is too long may cause latent actions to capture excessive visual changes, particularly in egocentric videos.
>
> ---
>
> **Q2. What is the purpose of learning the world model? It is entirely not used for low-level control.**
>
> In this work, the world model primarily serves to assess the semantic consistency of the learned latent actions across humans and robots.
>
> Besides, learning world models has long been a fundamental topic in embodied decision-making. By introducing a unified latent action space, the world model gains greater flexibility and potential. For future work, there is potential to extend the world model to synthesize data to address the data scarcity for policy learning.

---

> ### Author Response · Authors · 2024-11-26
> **We hope that our response addresses your concern**
>
> We sincerely appreciate the time you have taken to review our response. With our rebuttal now submitted, we are eager to know if it has adequately addressed your concerns. As the rebuttal phase draws to a close, we would be grateful for any additional feedback or clarification you might require.

---

> > ### Comment · Reviewer_68vo · 2024-11-28
> >
> > I thank the authors for their response. However, my major concerns regarding the lack of quantitative results and comparisons remain. Evaluating the method on only three in-distribution tasks in SIMPLER is insufficient to convincingly demonstrate its effectiveness. I strongly suggest expanding the variety of evaluation tasks. Additionally, I recommend including more comparisons with hierarchical planning approaches utilizing video or subgoal generation (such as SuSIE) or even simply training a state-of-the-art imitation learning policy (such as Diffusion Policy) on the action-full robot dataset. I would suspect that even the latter approach would perform well on SIMPLER's tasks, as the tasks are simple and in-distribution.

---

### Official Review · Reviewer_WjFr · 2024-11-08

**Soundness:** 1
**Presentation:** 2
**Contribution:** 1
**Rating:** 3
**Confidence:** 5

**Summary:**

The paper presents IGOR, a framework that learns a latent action space consistent between human and robot data. The latent action model can be used to train a foundation world model conditioned on latent actions. By labeling robotics dataset with latent actions and finetuning with real actions, IGOR enables training a foundation policy model.

**Strengths:**

- The paper seems technically correct, and experimental results are sound
- Other than the method section, the presentation is clear and text is well-written
- The idea of using image goal representation as latent action is, to my knowledge, new.

**Weaknesses:**

- It would help to add a method figure to visualize contents in section 2.1-2.3
- Missing citations, discussions, and baselines:
    - For latent action: various previous work has explored using point tracks (https://xingyu-lin.github.io/atm/, https://im-flow-act.github.io/) or text as latent action, how does IGOR compare to those, and why is image goal necessary given the alternative choices?
    - For world model: I would imagine if a VLM is used to convert a human video clip into a short text description, and then language-based video generation model can also generate robot videos. A similar idea using point tracks is also explored (https://homangab.github.io/gen2act/)
    - For policy model: prior works have explored vaiours alternative representations for latent actions: HOPMan (https://homangab.github.io/hopman/), Tract2Act (https://homangab.github.io/track2act/), Gen2Act (https://homangab.github.io/gen2act/) just to name a few. They have demonstrated more task diversity and difficulty than IGOR.
- As a foundation model, IGOR’s evaluations are too simple and not comprehensive enough
    - The paper could benefit from real robot experiments since IGOR policy is trained on a large-scale real robot dataset
    - The paper only demonstrates very simple 2-dimensional or gripper open/close tasks, which are not difficult from a robotics point of view, especially given how much data IGOR requires. Can IGOR handle more fine-grained or long-horizon tasks?
- Needs further clarification of contributions (see Questions section)

**Questions:**

What is the core contribution of IGOR? Is it the latent action model based on image and goal, or also the world model and policy model? Without judging its usefulness, I do believe the former is novel.  The claim for novelty of the latter seems unjustified without discussions or experimental justifications with regard to other baselines as I pointed out in the weakness section.

I also question the impact and utility of IGOR latent action model if it does not lead to a superior world model or policy model. The paper also doesn’t provide evidence that the IGOR latent action goes beyond coarse actions such as 2D movements and gripper open close, and language or point tracks may very well represent these actions well. Can IGOR represent actions that are more fine-grained (e.g. tasks demonstrated in the ALOHA paper), long-horizon? The authors are welcome to provide additional explanations or experimental results.

---

> ### Author Response · Authors · 2024-11-22
> **Response (2/2)**
>
> **Q4: Missing citations and discussions to related works**
>
> We really appreciate your comment for pointing out the missing related works. We have cited all these related works in Appendix E of our updated paper, and discuss the relationship of IGOR and these papers as follows.
>
> For latent action model and policy model related work: Any-Point Trajectory Modeling (ATM) and Tract2Act generates point tracking from the action-free videos for pretraining the policy to predict future trajectories of the tracked point, and learns a robot policy conditioned on generated trajectories. HOPMan generates future human plans as conditions for facilitating robot policy learning. Im2Flow2Act conditions the robot policy on complete generated object flows, which captures movement information only for the object, improving its cross-embodiment transfer capability.
>
> Same as ATM, HOPMan, Im2Flow2Act, and Tract2Act, IGOR also pretrain a foundation policy on action-free videos that generalize across embodiments. There are two major differences: (1) IGOR uses an unsupervised way to learn and compress the visual changes, while existing work needs an additional pretrained video model for point tracking; (2) IGOR uses a compact latent representation for visual changes, while existing works uses an explicit representation for visual changes. The compact action representation enables IGOR to transfer human actions on robots directly, as shown in Figure 2 in our paper.
>
> In the world model aspect, Gen2Act uses the VideoPoet model for text-to-video generation, and the generated human videos are used for downstream policy learning. Compared to existing world model works, IGOR can learn generalizable latent actions for fine-grained control on manipulation scenarios for both human and robots, while existing works use coarse grained text conditions for generating video predictions.
>
> ---
>
> **Q5: Add a figure for the method part (Section 2.1 - 2.3)**
>
> We really appreciate the comment, and have added a figure to describe our main method and network architecture in Appendix D of our updated paper.

---

> ### Author Response · Authors · 2024-11-22
> **Response (1/2)**
>
> We would like to thank reviewer WjFr for recognizing the novelty of our latent action approach, and for pointing out related works, such as point tracks or text as latent actions, which are alternatives of the latent action approach. We would love to address the reviewer’s questions in detail as follows.
>
> ---
>
> **Q1: Why is image goal (latent action) necessary given the alternative choices?**
>
> As mentioned in previous work that leverages actionless video data for robot learning, capturing motion in video is crucial, which is why there are numerous works on this topic. They employ different approaches such as point tracking, text, optical flow, and motion vectors from various research areas. While these methods are mature and deliver good performance, they all require additional prior knowledge from labeled information or labels generated by other models, falling under the category of supervised learning (SL) that leverages prior knowledge and structured data. In contrast, our latent action approach belongs to self-supervised learning (SSL), which has the potential to surpass supervised learning methods due to its ability to leverage large-scale unlabeled data.
>
> ---
>
> **Q2: Impact and utility of IGOR latent action model, and core contribution of IGOR**
>
> We propose the latent action approach as a self-supervised learning (SSL) alternative to existing methods like point tracking. Our efforts in this novel technical direction reveal the potential and possibilities of the latent action approach. Preliminary qualitative results demonstrate the semantic consistency of the learned unified action space across humans and robots using the world model. Additionally, preliminary quantitative results show the effectiveness of employing latent actions as sub-tasks within the policy model. Although these early results cannot yet match those achieved by more mature approaches in terms of task diversity and difficulty, we believe they demonstrate that the latent action approach is promising and merits further research.
>
> ---
>
> **Q3: Can IGOR represent actions that are more fine-grained (e.g. tasks demonstrated in the ALOHA paper) and long-horizon?**
>
> *More fine-grained latent actions*: We would like to provide more experimental results to show the expressiveness of IGOR latent action for capturing more fine-grained actions. We extract latent actions from out-of-distribution videos demonstrating various fine-grained control from Mobile ALOHA paper, and use the extracted latent actions and the initial frame to reconstruct the video by the IGOR world model. We provide the original video and generated video respectively in the “ALOHA Demonstration” part in the website [ https://iclr-2025-4517.github.io/index/rebuttal.html ]. The results indicate that IGOR can encode fine-grained robot controls in latent actions. We perform the same experiment with robot arm rotation, with the result shown in the “Latent Action for Rotation” part of the website [ https://iclr-2025-4517.github.io/index/rebuttal.html ]. The results show that IGOR can encode 3D robot arm rotations and movements in the latent actions.
>
> *Long-horizon*: Our existing latent action model captures the visual changes between two frames. If the frames are too far apart, the changes might be too large to model accurately, since it might be hard to find the same objects or scenes between the two frames. As a result, it would be better to have a sequence of latent actions to capture long-horizon visual changes.

---

> ### Author Response · Authors · 2024-11-26
> **We hope that our response addresses your concern**
>
> We sincerely appreciate the time you have taken to review our response. With our rebuttal now submitted, we are eager to know if it has adequately addressed your concerns. As the rebuttal phase draws to a close, we would be grateful for any additional feedback or clarification you might require.

---

> ### Comment · Reviewer_WjFr · 2024-11-28
> **Response to Author's Rebuttal of Q1**
>
> I understand the difference between SL and SSL outlined by the authors. I'd like to use the line of work that learns robotics knowledge from human videos (HOPMan, Track2Act, Gen2Act, and more recently ZeroMimic https://zeromimic.github.io/) as context for my discussion. It is true that these works use labels in the form of segmentation masks, point tracks, or hand reconstruction. But these labels are automatically generated by running existing off-the-shelf vision foundation models on unlabled datasets. Ultimately, the input and output of their systems are the same as this work: they also use unlabeled videos as input, and produce some "intermediate action representation" as output, with no extra information needed. I argue that the choice of producing this "intermediate action representation" using an off-the-shelf foundation that everyone has access to is a method choice and detail, and again does not change the input and output of the system. Therefore, I respectfully disagree with the authors' statement that the work "has the potential to surpass supervised learning methods due to its ability to leverage large-scale unlabeled data", given that these "supervised learning methods" in fact uses unlabeled datasets too.
>
> Each work chooses a different representation of action as their version of the "intermediate action": mask, point tracks, and in this work, some latent vector. Which action representation is better? This work provides no experimental evidence to show that their representation is more advantageous, only that it is "new", which I acknowledge. In fact, I would argue that the action representations in prior works have two obvious advantages over this work: 1. they are more interpretable, and 2. they are more scalable since they only need to train on human videos (note that in all the listed prior works, robot data is only used for downstream policy learning but not for training intermediate action generation), and also significantly less human videos data than this work, whereas this work requires training on both robot and human videos, making it less scalable.
>
>
> Finally, a note about using text labels. All of the datasets IGOR trains on actually have text labels, but it was the authors' choice not to use them. I would argue that using text to specify the task is more advantageous: it is much more intuitive and less cumbersome than using a goal image. Furthermore, using image goal to specify task is also not a new idea, and it has been explored in H2R (https://sites.google.com/view/human-0shot-robot), HOPMan, ZeroMimic, just to name a few.

---

> ### Comment · Reviewer_WjFr · 2024-11-28
> **Overall Reponse to Rebuttal**
>
> First, I'd like to apologize for my late response.
>
> Image goal representations are not a novel idea. As listed in my previous comment, a line of prior works (HOPMan, H2R, ZeroMimic, etc.) have explored generating intermediate actions from image goal representation by training on unlabeled datasets, same as IGOR. Latent actions as the intermediate action is indeed new, but there's no justification that this latent action is superior to other types of intermediate actions, therefore I don't believe this paper merits acceptance.
>
> Ultimately, I believe the best evaluation of an embodied AI method, whether it has potential and is promising, is with embodied AI performance itself, especially in comparison to prior methods. I am not convinced by the authors that IGOR has potential for embodied AI given its lackluster robot results and insufficient baseline comparisons.
>
> And therefore, I would like to keep my original score.

---

### Meta-Review · Area_Chair_FQD7 · 2024-12-19

**Metareview:**

**Summary**: This paper introduces IGOR (Image-GOal Representations), a framework that learns a unified, semantically consistent latent action space across human and robot activity data. IGOR uses latent actions derived from video data to train foundation policy and world models, which are then integrated with low-level policies for downstream robotic tasks. The framework enables the transfer of human behavior to robotic systems, demonstrated through tasks such as object manipulation.

**Strengths**:
- The idea of using image goal representation as latent action is novel.
- Several reviewers agree that the presentation is clear and well-written (for the most part).

**Weaknesses**:
- Simplistic evaluation: All reviewers noted that the evaluations are limited to relatively simple tasks in the SIMPLER simulator, with in-distribution tasks derived from the RT-1 dataset. Reviewer `68vo` emphasized the lack of out-of-distribution or real-world tasks, reducing confidence in the framework’s generalizability.
- Lack of strong baseline comparisons: Reviewers `68vo` and `WjFr` highlighted the lack of comparisons to more advanced methods, such as UniPi, SuSIE, or Gen2Act, which similarly leverage video data for policy learning. While the rebuttal included UniPi as a baseline, its integration was minimal, leaving gaps in assessing IGOR’s advantages.
- Scalability concerns: Reviewer `WjFr` questioned the scalability of the approach, particularly its reliance on both human and robot video data during training. Existing methods that utilize only human data (e.g., ZeroMimic) were argued to be more practical and scalable.

**Recommendation**: While IGOR presents an innovative framework with promising directions for embodied AI, the limited experimental scope, insufficient baseline comparisons, and scalability concerns significantly weaken its impact. The reviewers agree that the work demonstrates feasibility but lacks the empirical rigor and evidence needed for acceptance. As such, my vote is to Reject the paper.

**Additional Comments On Reviewer Discussion:**

The authors addressed several key concerns during the rebuttal:

- More baselines: UniPi was implemented as a baseline, with results indicating IGOR’s higher success rates. However, the reviewers found the scope of the comparison insufficient.
- Limited/simple experimental evals: The authors acknowledged the limitations of the SIMPLER tasks and emphasized that IGOR’s contributions lie in demonstrating feasibility. Future plans to explore more complex tasks were outlined but did not fully mitigate concerns.
- Overclaiming generalization/scalability: The authors clarified that IGOR’s latent actions offer compact representations suitable for fine-grained control, but reviewers remained unconvinced about their superiority over alternatives like point tracks or text-based actions.

Despite the authors' efforts, both `68vo` and `WjFr` retained their scores (3: reject), citing limited scope and insufficient evidence for IGOR’s broader claims. Unfortunately, I cannot accept the paper given these strong objections to its claims.

---

### Decision · Program_Chairs · 2025-01-22

Reject